# Fibre–Wood Laminate Biocomposites: Seawater Immersion Effects on Flexural and Low Energy Impact Properties

**DOI:** 10.3390/polym14194038

**Published:** 2022-09-27

**Authors:** Fabuer R. Valencia, Germán Castillo-López, Jon Aurrekoetxea, Alberto Lopez-Arraiza

**Affiliations:** 1Institut für Mechanik und Statik, Universität der Bundeswehr München, 85577 Neubiberg, Germany; 2Civil, Material and Manufacturing Engineering Department, Escuela de Ingenierías Industriales, University of Málaga, 29016 Málaga, Spain; 3Mechanical and Industrial Manufacturing Department, Mondragon Goi Eskola Politeknikoa, Mondragon Unibertsitatea, 20500 Arrasate, Spain; 4Faculty of Engineering in Bilbao, University of the Basque Country (UPV/EHU), 48920 Portugalete, Spain

**Keywords:** flax fibre, flax/wood hybrid, low energy impact, flexural, seawater ageing

## Abstract

The present paper explores a new concept of a hybrid eco-composite by substituting the natural fibre plies with thin wood veneers. The new composite, named Fibre–Wood Laminate (FWL), is inspired by fibre–metal laminate materials. The studied FWL configuration consisted of a single thin pinewood veneer at each of the outer layers of a flax woven fabric reinforced bio-epoxy composite manufactured by infusion. Three-point bending results showed that wood veneer gives a highly anisotropic nature to the FWL. In the best case, with the grain of the wood at 0°, the stiffness and the strength increased by 28 and 41%, respectively, but reduced the strain-at-break by 27% compared to the flax fibre reinforced bio-epoxy (FFRB). The penetration and perforation energy thresholds and the peak force of the FWL obtained by falling weight impact tests were 32, 29, and 31% lower than those of the FFRB, respectively. This weakening was due to using single wood veneers, so the challenge for improving impact properties will be to explore thicker FWLs with different stacking sequences and orientations. The effect of immersing the FWL in seawater also showed considerable differences. The epoxy matrix filled the cellular structure of the wood veneers, creating a barrier effect and reducing the amount of water absorbed by the flax fibres.

## 1. Introduction

Fibre reinforced polymers (FRPs) are becoming widely used in the manufacturing of products where a high mechanical property must be accompanied by a low weight. Consequently, these composite materials can be found in industrial sectors such as aeronautical, automotive, wind power, or naval [1,2]. The fibres are usually glass or carbon, while the polymer is usually an epoxy, vinylester, or polyester resin. However, there is a trend toward the replacing of synthetic fibres with natural fibres [3] and petroleum-based thermosetting resins with resins from renewable resources [4].

Natural fibres such as flax, rattan, bamboo, hemp, kenaf, or jute have high specific properties such as stiffness, impact resistance, and ductility. In addition, they are available in large amounts, and are renewable and biodegradable. Other desirable properties include low cost, low density, acoustic and thermal insulation, less equipment abrasion, less skin and respiratory irritation, good vibration damping, and enhanced energy recovery [5,6]. Among all the natural fibres, flax is a promising substitute for glass fibres for semi-structural and structural applications [7,8], with a lower environmental impact [9]. However, the hydrophilicity of the flax fibres results in high moisture absorption and weak adhesion to hydrophobic matrices [10]. 

Due to this hydrophilicity, the use of natural fibres in applications that are in contact with both freshwater and seawater, such as marine civil engineering constructions or most nautical structures, is limited [11,12,13,14,15,16]. In consequence, extensive studies have carried out hybridising flax fibres with synthetic fibres to achieve a better combination of mechanical properties [16,17,18,19,20,21,22,23,24]. Another approach for hybridising composites is using sheets instead of fibres, and Fibre–Metal Laminates (FML) have demonstrated good mechanical performances for marine applications [25,26,27]. Given that wood is a low-cost, eco-friendly, and traditional material used in marine constructions [28], similarly to FML, it can be integrated as veneer into biocomposites to obtain a Fibre–Wood Laminate (FWL) structure. Moreover, the position of the wood in the outer plies provides aesthetic properties to the laminate, highly valued in the nautical sector. According to an extensive literature search, to date, there are not any studies on the flax fibre/wood veneer hybrid reinforced bio-epoxy resins (FWL).

One of the most critical loads during its service life for a marine structure, such as a ship hull, is low-velocity impacts due to floating elements or manoeuvrers in the port [29,30,31]. Some research works have been specifically focused on understanding the impact damage response of flax fibre composites [16,17,18,19,20,21,22,23], but this field is still at its early stages of the investigation. Cuynet et al. [32] showed that even low energy impact events (5 J) could reduce the stiffness and strength of flax fibre reinforced polymers up to 10% and 20%, respectively. Therefore, their impact behaviour must be understood to be used in semi-structural and structural marine applications.

In an attempt to produce sustainable materials with high mechanical and impact performances, flax/pinewood hybrid laminates (FWL) are evaluated as reinforcements of a bio-epoxy resin. Quasi-static bending and low energy impact tests were performed before and after seawater immersion, and the damage extent was characterised by means of visual inspection. A flax fibre reinforced bio-epoxy (FFRB) composite was used as reference.

## 2. Materials and Methods

### 2.1. Materials and Manufacturing

The bio-epoxy used was SuperSap 100/1000 Entropy systemsupplied by Ferrer Dalmau, Barcelona, Spain. The mixing ratio of the INF02 curing agent was 100:33 by weight. The flax fibre reinforcement was supplied by Lineo Company as a bi-directional (0°/90°) woven fabric of 300 g/m^2^. The radiata pine tree wood was supplied by Maderas Lobera, Trapagaran, Bizkaia, Spain as veneers of 0.5 mm in thickness, close to the 0.75 mm of the flax fabric. Flax fabrics and wood veneers were conditioned at 50% relative humidity, suggested as the best conditions for obtaining balanced properties with flax fibres [33], and 23 °C for five days before manufacturing. Two different types of 300 mm × 300 mm laminates were manufactured by vacuum infusion:The flax fibre reinforced bio-epoxy laminates (FFRB) with five plies, given a thickness of 3.82 ± 0.07 mm and a 28.8 ± 0.12% fibre content in volume.Flax fibre/wood veneer laminates (FWL) of three flax fibre plies in the core and a single pinewood in each of the outer layers (Figure 1a). The final nominal thickness was 3.2 ± 0.03 mm, and the fibre content was 30.6 ± 0.12% in volume.

Infusing FWLs consists of drilling several holes at the lateral side of the wood veneers to facilitate the resin flux in the *z*-axis. In addition, a net bleeder over the peel ply was placed to facilitate the resin distribution in the x-y plane (Figure 1b). Vacuum infusion was carried out at room temperature, followed by a post-cure at 80 °C for 8 h. The quality of the FRRB and FWL laminates was inspected as manufactured by C-Scan. The equipment used consists of an OmniScan MXU (Olympus, Tokyo, Japan) portable flaw detector, two-axis GLIDER™ scanner, broadband phase array probe of 1 MHz, and specific wedges, showing that they were free of delamination or dry zones. Void content measured according to ASTM D2734-16 was, in all cases, lower than 4%.

### 2.2. Seawater Immersion

Samples were immersed in the 20 m^3^ tank of the Research Centre in Experimental Marine Biology and Biotechnology (PIE) (Basque Country, Spain) at the Cantabrian Sea. The immersion period was from April to October, the period of higher marine bioactivity, longer than the two months reported in [15] for saturating the FFRP composites. The tank seawater was renewed continuously with 300,000 L/day at an average temperature of 21 ± 3 °C. The samples were clamped into a square frame, avoiding the contact of the edges with the seawater and consequently the fast saturation due to the exposed fibre ends [13]. Furthermore, this setup reproduces the actual situation of the case of a ship hull. The time elapsed between the extraction of the specimens from the seawater tank and the mechanical tests was always less than 48 h.

### 2.3. Three-Point Bending Tests

Flexural tests were conducted in accordance with ISO 14125. A universal test machine Servosis ME 405 (Servosis, Madrid, Spain) equipped with an HBM S9M/5 kN (in a 1 kN scale) was used. A 60 mm span and a crosshead of 2 mm/min were used. Five samples of each laminate were tested before and after seawater immersion. The specimens were machined to their nominal dimensions 15 mm × 90 mm. Regarding the flax fibre/wood veneer hybrid biocomposites, the specimens were machined both in the direction of the wood grain (FWL0) and in the perpendicular direction (FWL90).

### 2.4. Low-Velocity Impact Tests

Low-velocity impact tests with different impact energies were carried out at room temperature on FFRB and FWL specimens. The drop-weight machine, a Fractovis-Plus from Ceast (Pianezza, Italy) was equipped with a 20 kN load cell attached to a 20 mm diameter hemispherical impact tup. Combining different impact masses and falling heights (2.045 to 3.045 kg and 0.05 to 1 m) enabled an energy (*E*_0_) range between 1 and 30 J, covering a whole range of behaviours up to the perforation of the samples. The anti-rebound system of the machine avoided second impacts in rebound scenarios. The 70 mm square plates were placed on a ring support fixture with an inner and outer diameter of 40 mm and 60 mm, respectively. Additionally, they were clamped by a pneumatic device consisting of another annular tool.

Taking Newton’s second law and the contact force–time data (*F*(*t*)) recorded during the impact events as the starting point, displacement–time (*δ*(*t*)) and energy–time (*E*(*t*)) curves were calculated by integration [34]. Figure 2 shows the most relevant impact properties of the composites identified from the *F*(*t*) and *E*(*t*) impact curves. The damage energy threshold (*E*_d_), which corresponds to the first inflection point of the force–time curve (*F*_d_), divides the impact events into two categories. Depending on whether the incident energy is below or above *E*_d_, impact events are classified as subcritical or supercritical. The peak force (*F*_p_) and dissipated energy (*E*_dis_) are also relevant features when characterising the impact behaviour of composite materials.

### 2.5. Scanning Electron Microscopy

The failure surfaces of flexural for specimens were used to illustrate the adhesion between the resin and the flax fibre/pinewood laminates. This analysis was performed using a Jeol JSM-6400 (JEOL, Tokyo, Japan) Scanning Electron Microscope (SEM). Fracture surfaces of the composite samples were coated with gold and then analysed using the SEM operated at 20 kV.

## 3. Results and Discussion

### 3.1. Seawater Immersion

All the biocomposites experienced discolouration after the long-term immersion, showing that the absorbed water molecules in the bio-epoxy matrix accelerate the photo-oxidation reactions [35]. This was after six months of seawater immersion, enough for saturating the biocomposites, as reported by [15]. Consequently, the FFRB thickened by 2.6%, whereas the wood veneer acted as a barrier against the water absorption of the flax fibres, and the FWL did not show a remarkable increase in thickness.

### 3.2. Three-Point Bending Properties

Table 1 shows the flexural test results of both the FFRB and FWL before and after seawater immersion. Regarding the FWL, specimens machined in the direction of the wood grain (FWL0) and across it (FWL90) were tested. The values of the FFRB without immersion are similar to those in the literature [11,12]. Comparing the results of the FWL0 with those of the FFRB, the pine tree wood veneers increased flexural modulus (27%) and strength (40%), whereas they reduced the elongation-at-break by 25%. In contrast, FWL90 showed the lowest flexural properties due to the perpendicular direction of the wood grain. Thus, the highly anisotropic nature of the veneer can be beneficial if the grain direction is suitably oriented parallel to the main stress direction.

Figure 3 shows the variation in bending properties of the three biocomposites after six months of immersion in seawater. The degradation process reduced the stiffness of all the biocomposites, to a greater extent in the FFRB (23.5%), followed by FWL0 (11.5%) and FWL90 (9.8%). The lost strength was practically the same, about 11%, for the three biocomposites. The higher divergence was found in elongation-at-break, since the FFRB ductility increased by 12.1%, whereas FWL0 and FWL90 decreased by 8.3% and 11.1%, respectively. The effects of seawater immersion of the FFRB were similar in trend and quantitatively to those reported in the literature [11,12]. Thus, the degradation of flax fibres, the epoxy matrix, and the bond at the fibre/matrix interfaces after the immersion reported in those studies should be at the origin of the variations in the FFRB behaviour. Regarding the FWLs, wood veneers protected the inner flax fibres, reducing their water absorption and the consequences in the flexural properties.

From a structural design point of view, the results of the three-point bending tests show that designers must especially consider the reduction in stiffness when immersing the FFRB in seawater. In the case of wet FWLs, the loss of stiffness and strength is of equal importance.

### 3.3. SEM Analysis

Fractography analysis of failure surfaces at different magnifications helps to understand three-point bending results. Before seawater immersion (Figure 4a), the failure mode of the FFRB is typical of brittle materials. The ellipse in Figure 4a shows the detachment of the flax fibre due to the low fibre–matrix adhesion, usual in polymeric composites based on flax fibres [11,12]. After seawater immersion (Figure 4b), the flax fibres grow in volume due to their hydrophilicity, leading to a stronger fibre–matrix physical adhesion. The specimen broke by fibre pull-out and fibre/matrix debonding, indicating that the fibre/matrix interface damage could be due to the higher strain-at-break of the wet FFRB.

Figure 5 shows the fracture surfaces of FWL0 before (a) and after (b) seawater immersion, which are also representative of FWL90. The first relevant conclusion is that the epoxy matrix filled the cellular microstructure of the wood during the infusion manufacturing process, marked with an ellipse in Figure 5a. There is a physical connection between epoxy and wood cell walls, which can explain the barrier effect against water absorption in the FWLs. Regarding the fracture micromechanisms, on the one hand, the cell walls of the pinewood veneer showed brittle fracture after immersion (Figure 5b). On the other hand, the failure of the flax fibre yarns along the load direction showed debonding, fibre pull-out, and brittle fracture. The epoxy matrix showed a smooth surface in both cases, typical of brittle fracture.

### 3.4. Low-Velocity Impact Tests

The energy plot represents the evolution of the dissipated energy (*E*_dis_) versus the incident impact energy (*E*_0_). The equal energy line (dashed line) is plotted as a diagonal in the energy plot and identifies the upper limit of *E*_dis_. Figure 6 shows the energy plots for the FFRB and FWL composites. At dry conditions (Figure 6), both materials have three regions. At the lower incident energy levels, *E*_dis_ increased quadratically up to the equal energy line, which defines the penetration energy threshold. This region was composed of two behaviours: subcritical and supercritical before penetration. However, the limit between them (*E*_d_) was not identifiable in this plot. The quantitative analysis of this first region shows that the FWL dissipated more energy than the FFRB. Nevertheless, it implies a more severe damage level of the FWL [34] and a lower penetration threshold of 12.5 J, whereas FRRB still rebounded the impact up to 18.5 J. In the penetration region, the second one, composites dissipated all the incident energy, but the impactor could not penetrate the sample. The energy range in this region was small for both materials, since the perforation threshold for the FWL and FRRB was 13.4 and 18.8 J, respectively. This perforation energy represents the maximum capacity of the material for dissipating impact energy and reveals that hybridising the FRRB with the wood veneers is not recommended when the application requires to maximise the dissipation of impact energy. This effect is associated with the fact that under impact loading of a single thin veneer, shear failure parallel to the grain direction of the wood plays a weakening role [36].

Energy plots for the wet FRRB and FWL composites (Figure 7), as for the dry composites, show the three regions, but penetration and perforation thresholds are higher. The FRRB critical energies increase to 22.3 and 24.8 J, respectively, while for the FWL, 14.6 J is needed for penetrating and 15.1 J for perforating. The origin of this improvement could be attributed to variations in the matrix and reinforced behaviour when immersed in seawater. The absorbed water by the flax fibres and the wood veneer softens them [11,16,20], and even if it is secondary, the matrix [14,19], leading to higher strain before failure. Additionally, flax fibres swell with absorbed water molecules, which could fill the gaps between the fibre/matrix interface [11] and between the fibres in the bundle [19], increasing the interfacial strength. The fact that the penetration and perforation thresholds of the wet FWL increased less than in the FFRB case seems to be related to the barrier effect that can play the wood veneer, reducing the amount of water absorbed by the flax fibres.

The visual inspection of the damage on the impacted and back faces supports the conclusions of the energy plots related to the penetration and perforation thresholds. As can be seen in Figure 8a, the dry and wet FFRBs were not perforated at 10 J. On the impacted face of both composites, the hemispheric impactor left a permanent dent. In the dry FFRB sample, the impactor induced concentric rings of damage, while on the wet FFRB it did not. On the back face of the dry FFRB, fibre breakage-dominated micromechanisms [37] generated a cross-shaped cracking, with the cracks parallel to the warp and weft directions of the fabric. Instead, on the wet sample (Figure 8b), only plastic deformation developed at the dent. Therefore, it is demonstrated that the higher energy dissipated by the dry FFRB was at the cost of more damage. The images of the samples impacted at 20 J (Figure 8c) also show a significant modification of the failure induced by the water absorption. Whereas the dry FFRB was completely perforated, with matrix and fibre fracture along the principal fibre directions and flexural bending failures of the quadrants created, the wet FFRB was still in the penetration energy region (Figure 8d). Thus, the plasticisation and flax fibre swelling effects of water absorption improve the maximum energy dissipation capacity of the FFRB composites.

The images of impacted FWL also help to understand the results of the energy plot. In the 10 J impact tests, the FWL did not perforate, neither in its dry nor wet condition. At the front face of the dry FWL impacted at 10 J (Figure 9a), the wood veneer was fractured at the contour of the dent. The back face damage consisted of a cross-shaped cracking mechanism, with a longer crack length parallel to the wood grain than in the perpendicular direction. The damage mechanisms in the wet FML (Figure 9b) were less severe, as in the front face the wood veneer was not fractured yet, and in the back the cross-shaped cracking was in its earliest stage. The images from the FWL impacted at 15 J show that the dry FWL was perforated (Figure 9c), with almost no damage abroad the impactor contact zone, whereas a depth dent, without perforation, can be seen in the wet FWL (Figure 9d). As in the FFRB composite, in the FWL, the absorbed water reduced its damage and was at the origin of the moderate energy dissipation improvement.

The peak force versus the incident energy curves, such as those for the FFRB and FWL composites in Figure 10, increase according to a power law and reach a plateau associated with the maximum allowable impact load of the material [34]. The first conclusion is that the hybridisation with a wood veneer does not improve the maximal force, as the FFRB reached a value of 2798 ± 69 N, whereas the FWL only withstood 2232 ± 86 N. The second conclusion is that water absorption did not improve the peak force values. Indeed, the maximum load for the wet FFRB was practically identical (2911 ± 11 N) and slightly lower for the wet FWL (1986 ± 31 N).

Force–time and energy–time impact curves (Figure 11) add information to understand the difference in behaviour shown by the FFB and FWL in dry and wet conditions. Regarding the supercritical impacts before penetration (10 J), the dry and wet FFRBs showed almost the same curves (Figure 11a), with the damage threshold approximately located at 1620 N, corresponding to 1.3 J. Following the damage threshold, the force stabilised in a plateau and increased up to the peak force. Finally, when the unloading path started, the contact force and energy reached their maximum value, and the impactor rebounded, as can be deduced from the energy reduction. The 10 J impact curves for the dry and wet FWLs (Figure 11b) were similar but quite different to those of the FFRB. The inflexion point of damage threshold for the wet FWL was slightly lower, located at 1100 N and 0.7 J, whereas the damage threshold for the dry FWL was at 1480 N and 1.15 J. As with the penetration and perforation thresholds, the FWL’s damage threshold was lower than the FFRB’s. The force increased continuously after the damage threshold up to the peak force. The main difference concerning the FFRB was in the post-peak region, as damage developed at an almost constant force level in the FWL and rebound did not start at the peak force.

The perforation impacts at 30 J for the FFRB (Figure 11c) and 20 J for FWL (Figure 11d) revealed the most relevant effects of the absorbed water on the impact behaviour of the biocomposites. Starting with the FFRB, increasing impact energy did not modify the damage threshold. The contact force increased up to the same peak value in the case of the dry and wet FFRBs. Behaviour differed after the peak force; in the dry FFRB, the contact force dropped suddenly in three stair-like paths, whereas in the wet FFRB, it kept at the maximum during a short contact time and dropped continuously. The shorter total contact time of the dry FFRB also supported the fact that it was more brittle. Thus, the higher energy dissipated by the wet FFRB seems to be due to the damage propagation mechanism, which is directly related to the fracture energy. 

The results reported by Todo et al. [38] demonstrated that dynamic interlaminar fracture toughness of carbon/epoxy composites could increase with water absorption, supporting the energy dissipated capacity improvement of the wet FFRB. Furthermore, in the presence of hydrophilic flax fibres, toughening of the matrix could increase even more [39].

The perforation of the FWLs took place at lower impact energies (Figure 11d). Like in the FFRB, in the dry and wet FWLs, the initial regions of the force–time curves, up to the damage threshold, were identical to those obtained at 10 J. The peaks for both the FWLs were at the same force value and located at the same contact time. As in the FFRB, the differences were in the post-peak region, where the wet FWL showed a less pronounced decreasing trend and consequently dissipated more energy. In the FWL case, the higher fracture toughness associated with the water absorption [38,39] also can justify the higher dissipated energy, but to a lower extent than in the FFRB since the FWL absorbed less water.

## 4. Conclusions

This paper aimed to deduce whether the hybridisation of flax fibre reinforced biocomposites with thin pinewood veneers is beneficial in terms of quasi-static flexural and impact properties. Furthermore, this paper aimed to determine the effect of seawater immersion on the properties of the biocomposites under study. Based on the experimental results, one can conclude that:The results of the three-point bending tests showed that hybridisation of the FRPP with thin wood veneers confers a highly anisotropic character to the resulting FWL. Thus, hybridisation could be beneficial in terms of stiffness and strength, as long as the wood grain and principal stress directions are aligned parallel to each other.During the infusion, the epoxy matrix filled the cellular structure of the wood, which acted as a barrier and reduced the water absorbed by the flax fibres.The protection of this matrix-impregnated wood veneer reduced the stiffness loss in seawater immersion situations, but the strength reduction is the same as in the net FFRB.Regarding the impact results, the wood veneer had a negative effect, since it reduced the penetration and perforation energy thresholds. The unidirectional wood grain of the thin veneer integrated into the FWL was at the origin of this weakening, so introducing several thin wood veneers with different grain orientations could improve the impact performances of the FWL. Thus, the FWL concept should be explored in thicker laminates with a higher amount of wood veneers and with different ply configurations.The impact results of the wet FFRB and FWL showed a trend opposite to the three-point bending quasi-statics, as the critical penetration and perforation energies increased. The improvement was noticeable for the FFRB and almost negligible for the FWL. Therefore, the seawater protective effect of the wood veneers was detrimental in the case of the FWL.

Based on the conclusions of the present research work, one could state that FWLs are challenging materials for applications in contact with seawater. However, new investigations focused on thicker laminates with different wood veneer directions and stacking sequences should be carried out to profit from the advantages of FWLs.

## Figures and Tables

**Figure 1 polymers-14-04038-f001:**
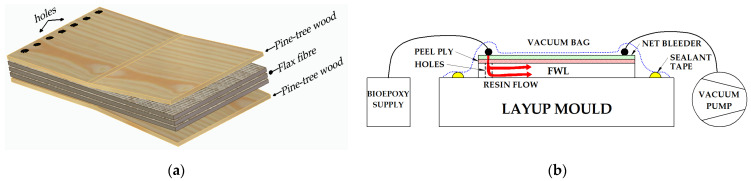
(**a**) Stacking configuration of FWL; (**b**) schematic diagram of vacuum resin infusion process for FWL.

**Figure 2 polymers-14-04038-f002:**
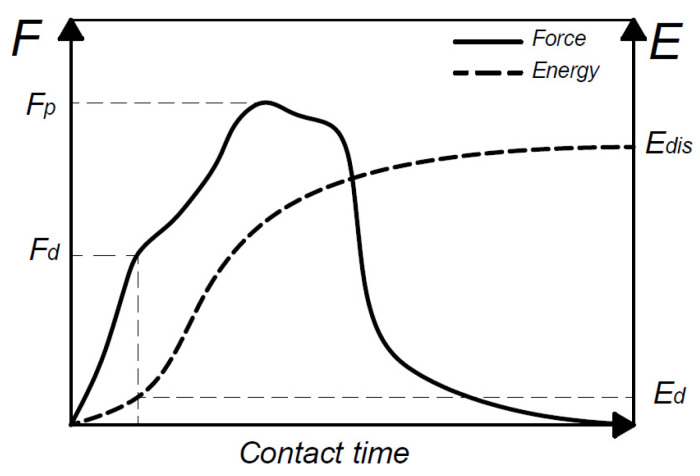
Schematic diagram of typical force and energy vs. time impact curves for composites.

**Figure 3 polymers-14-04038-f003:**
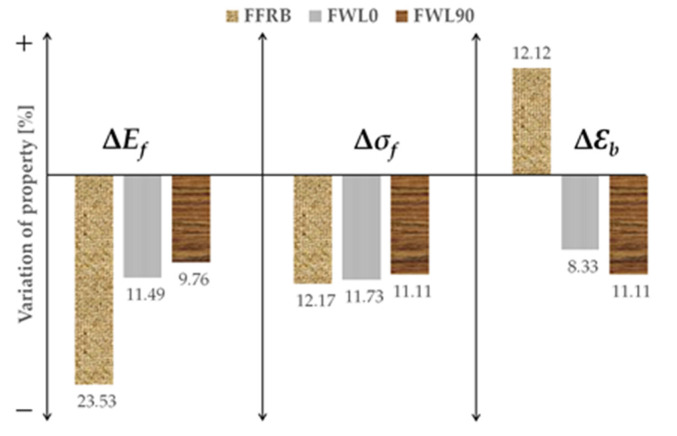
Variation in flexural properties after seawater immersion.

**Figure 4 polymers-14-04038-f004:**
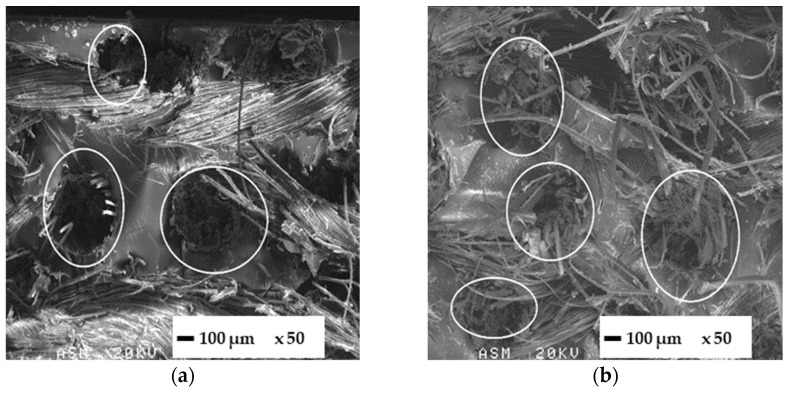
SEM micrographs of the FFRB (×150) before (**a**) and after (**b**) seawater immersion.

**Figure 5 polymers-14-04038-f005:**
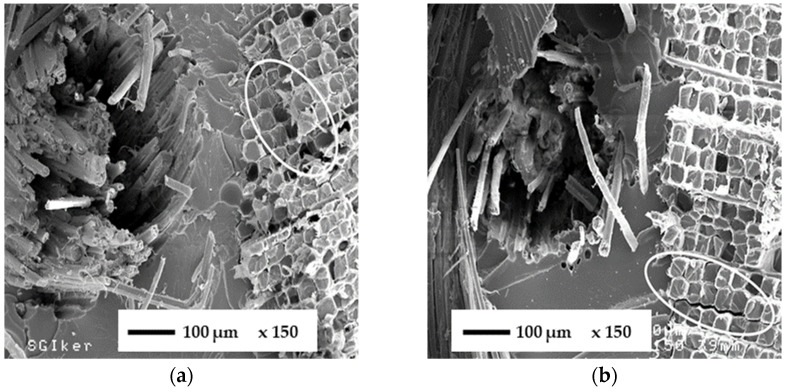
Fracture surface micrographs of the FWL0 (×150) before (**a**) and after (**b**) seawater immersion.

**Figure 6 polymers-14-04038-f006:**
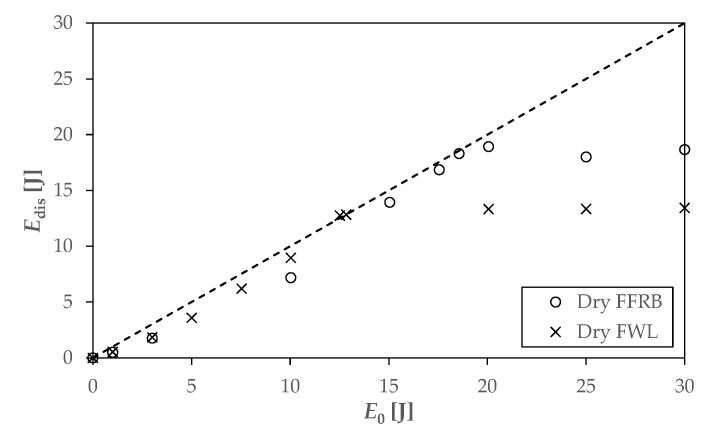
Energy plots of the dry FRRB and FWL composites.

**Figure 7 polymers-14-04038-f007:**
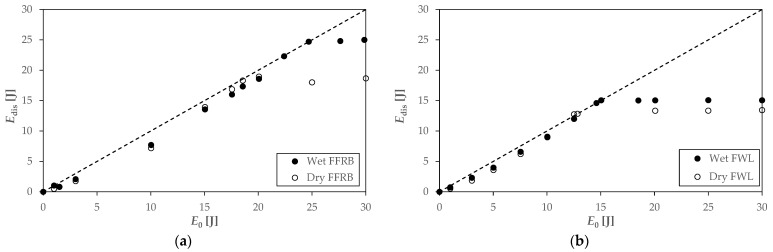
Energy plots of the wet FRRB (**a**) and FWL composites (**b**).

**Figure 8 polymers-14-04038-f008:**
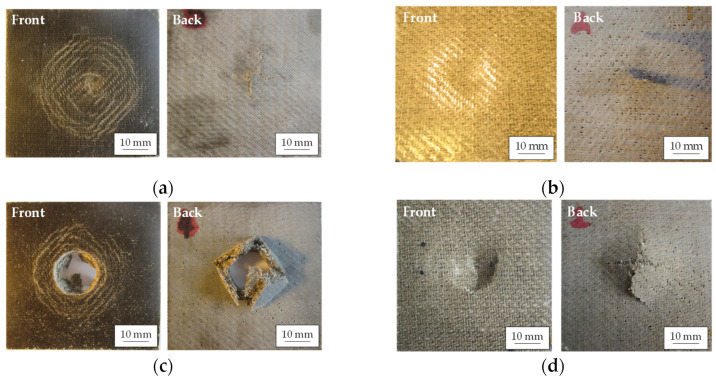
Representative post-impact images of the front and back faces of dry FFRB at 10 J (**a**), wet FFRB at 10 J (**b**), dry FFRB at 20 J (**c**), and wet FFRB at 20 J (**d**).

**Figure 9 polymers-14-04038-f009:**
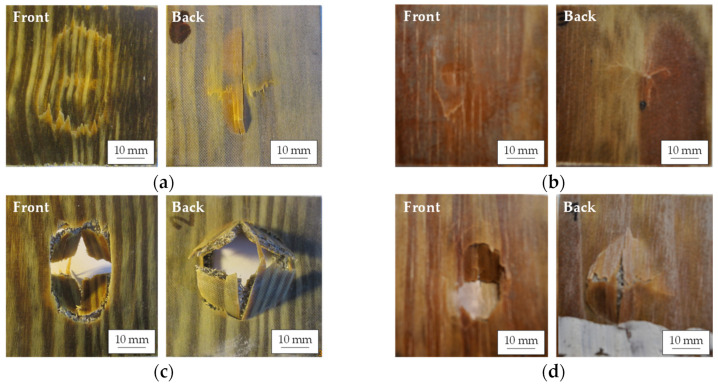
Representative post-impact images of the front and back faces of dry FWL at 10 J (**a**), wet FWL at 10 J (**b**), dry FWL at 15 J (**c**), and wet FWL at 15 J (**d**).

**Figure 10 polymers-14-04038-f010:**
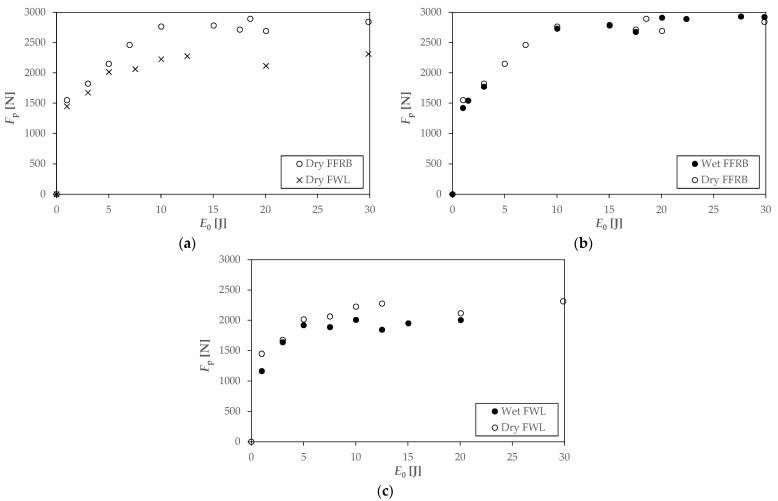
Peak force plots of the dry FRRB and FWL (**a**), wet FFRB (**b**), and wet FWL (**c**).

**Figure 11 polymers-14-04038-f011:**
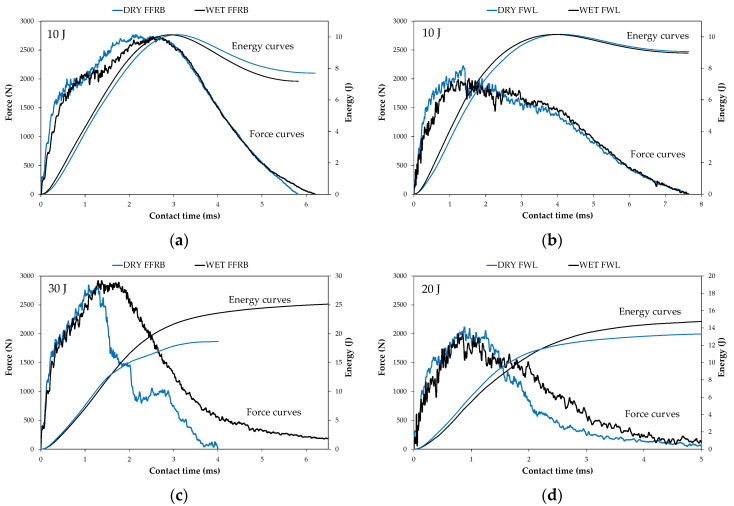
Force-time and energy-time curves before penetration for dry and wet FFRBs (**a**), before penetration for dry and wet FWLs (**b**), perforation for dry and wet FFRBs (**c**), and perforation for dry and wet FWLs (**d**).

**Table 1 polymers-14-04038-t001:** Flexural test results.

Material	Before Seawater Immersion			After Seawater Immersion		
	*E_f_* (GPa)	*σ_f_* (MPa)	*ε_b_* (%)	*E_f_* (GPa)	*σ_f_* (MPa)	*ε_b_* (%)
FFRB	6.8 ± 1.1	115 ± 5.0	3.3 ± 0.1	5.2 ± 0.6	101 ± 5.0	3,7 ± 0.1
FWL0	8.7 ± 0.5	162 ± 4.9	2.4 ± 0.1	7.7 ± 0.1	143 ± 1.8	2.2 ± 0.3
FWL90	4.1 ± 0.6	81 ± 4.0	3.6 ± 0.1	3.7 ± 0.5	72 ± 4.0	3.2 ± 0.1

*E_f_*: Flexural modulus; *σ_f_*: flexural strength; *ε_b_*: elongation-at-break.

## Data Availability

No new data were created or analysed in this study. Data sharing is not applicable to this article.

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
