# Peer review of "Fibre–Wood Laminate Biocomposites: Seawater Immersion Effects on Flexural and Low Energy Impact Properties"

_polymers, 2022, doi:10.3390/polym14194038_

Round 1
Reviewer 1 Report
The research content of this article is meaningful, but more experimental information is needed.
1. Authors should provide photographs or sketches of each step of the material preparation process.
2. The experimental setup is also very important information. In addition to the written description, the author should provide the experimental setup that annotates the complete experimental equipment and experimental process.
3. The time factor should be one of the most important factors and variables in the study of the influence of seawater infiltration effect. The author only made one group of infiltration time, and if several groups of experiments with different infiltration time can be added to get a group of variable curves with time, the study will be more perfect.
4. Studies have shown that some of the performance decreases after prolonged infiltration, and these data are best quantified using proportional data, such as percentage reduction.
5. The author has to read through the article again to make sure there are no low-level errors, like 145 lines of immersion written as inmersion
Reviewer 2 Report
Comments
In this paper, the effects of seawater immersion on the quasi-static bending and low energy impact properties of flax/pinewood hybrid laminates (FWL) were studied. In general, the work is well done and the conclusion is supported by the experimental and results. However, there are still some issues to be addressed before its acceptance.
Abstract:
1. The author said “in the best case, the stiffness and the strength increased by ……., by 27%.” Is this an occasional data? An average data should be provided in this part and standard error also need to be presented.
2. The penetration and perforation energy thresholds and the peak force of the FWL obtained by falling weight impact tests were 32, 29 and 31% lower, respectively. Above data is good or bad? A comparison with other materials is encouraged.
Introduction
1. The author said “Natural fibres such as flax, bamboo, hemp, kenaf or jute have high specific properties such as stiffness, impact resistance and ductility; Other desirable properties include low cost, low density, acoustic and thermal insulation, less equipment abrasion, less skin and respiratory irritation, good vibration damping, and enhanced energy recovery [5]” Rattan is also a kind of important biomass resources, which is third in abundance to timber and bamboo. It should be mentioned in this part. Some references about rattan need to be cited, such as "Top-down" fabrication of anisotropic, lightweight, super-amphiphobic, and thermal insulating rattan aerogels; Extraction and characterization of novel ultrastrong and tough natural cellulosic fiber bundles from manau rattan (Calamus manan); Micro- and nano-fibrils of manau rattan and solvent-exchange-induced high-haze transparent holocellulose nanofibril film (Please cite this article as: X. Han, J. Wang, J. Wang, et al., Micro- and nano-fibrils of manau rattan and solvent-exchange-induced high-haze transparent holocellulose nanofibril film, Carbohydrate Polymers (2022), https://doi.org/10.1016/j.carbpol.2022.120075)
Results and Discussion
1. Table 1, the symbol of elongation at break after seawater immersion is wrong.
2. In Figure 5, the author said the epoxy matrix filled the cellular microstructure of the wood. Please mark the epoxy resin in Figure 5. In addition, the author should describe the connection type between epoxy and wood cell walls, physical fill or chemical bonding?
3. In Figure 5 result part, the flax looks having a big width. How did it is immersed into wood cellular microstructure?
4. Figure 11 can show DRY FFRB with more vivid colors, which makes the image easier to recognize.
Round 2
Reviewer 1 Report
The manuscript has been revised accordingly. It is recommended to be published.